# Comparative Study of the Aftereffect of CO_2_ Inhalation or Tiletamine–Zolazepam–Xylazine Anesthesia on Laboratory Outbred Rats and Mice

**DOI:** 10.3390/biomedicines10020512

**Published:** 2022-02-21

**Authors:** Oksana N. Khokhlova, Natalya A. Borozdina, Elena S. Sadovnikova, Irina A. Pakhomova, Pavel A. Rudenko, Yuliya V. Korolkova, Sergey A. Kozlov, Igor A. Dyachenko

**Affiliations:** 1Biological Testing Laboratory, Branch of Shemyakin-Ovchinnikov Institute of Bioorganic Chemistry, Russian Academy of Sciences (BIBCh RAS), 6 Prospekt Nauki, 142290 Pushchino, Russia; khohlova@bibch.ru (O.N.K.); sadovnikova@bibch.ru (E.S.S.); pakhomova@bibch.ru (I.A.P.); pavelrudenko@bibch.ru (P.A.R.); dyachenko@bibch.ru (I.A.D.); 2Department of Molecular Neurobiology, Shemyakin-Ovchinnikov Institute of Bioorganic Chemistry, Russian Academy of Sciences (IBCh RAS), 16/10 Miklukho-Maklay Str., 117997 Moscow, Russia; serg@ibch.ru

**Keywords:** laboratory rodents, Sprague Dawley rats, CD-1 mice, serum biochemistry, animal care and protection

## Abstract

CO_2_ inhalation is currently the most common method of euthanasia for laboratory rats and mice, and it is often used for further terminal blood sampling for clinical biochemical assays. Lately, this method has been criticized due to animal welfare issues associated with some processes that develop after CO_2_ inhalation. The stress reaction and the value of the clinical laboratory parameters significantly depend on the used anesthetics, method, and the site of blood sampling. Especially in small rodents, an acute terminal state followed by a cascade of metabolic reactions that can affect the studied biochemical profile may develop and cause unnecessary suffering of animals. The aim of this study was to compare the stability of biochemical parameters of outbred Sprague Dawley rats and CD-1 mice serum collected after CO_2_ inhalation or the intramuscular injection of tiletamine–zolazepam–xylazine (TZX). The serum content of total protein and albumin, cholesterol, triglycerides, aspartate aminotransferase (AST), alanine aminotr ansferase (ALT), alkaline phosphatase (ALP), total bilirubin, and creatinine was decreased by the injection of TZX in comparison with CO_2_ inhalation. In addition, the levels of calcium, phosphates, chlorides and potassium were lowered by TZX vs. CO_2_ administration, while the level of sodium increased. Finally, the level of the majority of serum clinical biochemical parameters in rats and mice tend to be overestimated after CO_2_ inhalation, which may lead to masking the possible effect of anti-inflammatory drugs in animal tests. Injection anesthesia for small rodents with TZX is a more feasible method for terminal blood sampling, which also reduces the suffering of animals.

## 1. Introduction

Clinical biochemical blood assay is often an obligatory part of toxicological safety studies performed on laboratory outbred rats and mice at the terminal stage of the study. Currently, outbred rodents occupy a leading position compared to isogenic animals in toxicological and drug activity studies. This is due to the fact that outbred mice are more accessible and prolific; similarly, preclinical studies with outbred rats are published more often than isogenic ones [1,2,3,4,5]. In small rodents, the required volume of blood for the extended biochemical assay in toxicological studies can often be sampled only at the terminal stage of the study, when the animal is anesthetized or euthanized. It is known that the stress reaction and the value of the clinical laboratory parameters significantly depend on the used anesthetics, method, and site of blood sampling [6,7,8,9]. CO_2_ inhalation is one of the recommended methods of euthanasia in rats and mice [10], which is also widely used instead of general anesthesia for terminal blood sampling from the heart or vena cava inferior. However, it should be noted that at the recommended procedure of CO_2_ inhalation in a chamber [10,11], small rodents develop a terminal state accompanied by hypercapnia, acidemia and hypoxia in tissues [12,13], which should be taken into account during the assay of serums biochemical parameters. For animals used in experiments for inflammation level estimation or tissue integrity, for example, with anti-inflammatory drug tests, the choice of method of terminal blood sampling for the clinical biochemistry becomes significant.

Lately, CO_2_ inhalation as a method of euthanasia has been widely discussed in terms of animal welfare [14]. According to some data, CO_2_ inhalation leads to the activation of hypothalamic–pituitary–adrenal axis, with a further cascade of stress biochemical reactions [15,16,17]. When using another volatile agent, isoflurane, the animal recovers from anesthesia less predictably and faster (about 2 min after inhalation stops), which makes this method less acceptable due to the possibility of the animal awakening before or at the time of blood sampling. One of the alternative methods for the euthanasia of rats and mice for the biochemical blood sampling can be injectable anesthesia. The anesthesia for laboratory animals is performed by the combination of dissociative anesthetic ketamine with α2-agonists medetomidine, and dexmedetomidine or xylazine. However, earlier, it was shown that rodents who received anesthesia with ketamine combined with α2-agonists had an elevated level of AST, ALT and creatine kinase [18]. It is also known that the combination of ketamine–xylazine in animals provokes the development of hyperglycemia due to the activation of glycogenolysis and the inhibition of insulin expression [19,20,21]. Glycogenolysis does not develop in rats receiving a mixture of tiletamine–zolazepam-xylazine (TZX) [20]. Anesthesia for large animals is successfully performed with another dissociative anesthetic tiletamine combined with zolazepam and butorphanol [22] or xylazine [23,24,25].

In a study on the blood serum of intact rats anesthetized with TZX mixture, the levels of interleukin 1 beta (IL-1β) and tumor necrosis factor alpha (TNF-α) were barely determined [26], and TZX anesthesia is already used in animal models of allergic reactions [27]. On the contrary, the usage of ketamine–xylazine anesthesia increases the expression of IL-1ß and interleukin 6 (IL-6), and reduces the expression of TNF-α in rats [28], which may negatively affect the results’ validity. There is no unambiguous data about the effects of TZX mixture on blood composition yet, although blood parameter is one of the mandatory analyses in many projects of anti-inflammation, autoimmune, and toxicological studies. We compared the clinical biochemical parameters of the serum taken from the male and female outbred Sprague Dawley rats and outbred CD-1 mice after anesthesia with TZX or CO_2_ inhalation. The obtained results showed that anesthesia of small rodents with TZX is the most suitable method for terminal blood sampling.

## 2. Materials and Methods

The study was performed by the Biological Testing Laboratory in the Branch of Shemyakin-Ovchinnikov Institute of Bioorganic Chemistry of the Russian Academy of Sciences, according to the Guide for Care and Use of Laboratory Animals (8th edition, Institute for Laboratory Animal Research). The laboratory is accredited by AAALAC International. The procedures for animals were approved by the Institutional Animal Care and Use Committee. Specific pathogen free (SPF) outbred Sprague Dawley rats and CD-1 mice were received from the laboratory of animal breeding center “Puschino” at the age of 6–7 weeks and acclimatized for 2–3 weeks before the experimental procedures in the rooms for animals with the controlled environmental conditions: air temperature 20–24 °C, relative humidity 30–60%, 12-h cycle of daylight and at least ten-fold change of room air per hour. Five animals were kept in each standard polycarbonate Type 3 cage (820 cm^2^, mice) or Type 4 (1450 cm^2^, rats) with bedding material (LIGNOCEL BK 8/15, JRS, Rosenberg, Germany), free access to feed (SSNIFF V1534-300, Spezialdiaeten, GmbH) and filtered tap water. The cages with rats were provided with wood fibers for nesting (LIGNOCEL^®^, JRS, Rosenberg, Germany), and the cages with mice contained Mouse House™ (Techniplast, Buguggiate, Italy). Before blood sampling, the animals were deprived of food: rats for 16–18 h at night and mice for 4 h in the morning.

Female rats, male rats, female mice and male mice were divided into two groups of 10 animals. Such group size was set up after the sample size power calculation (http://www.biomath.info/power/index.html, accessed on 21 December 2019) at 80% power and 5% alpha values. Before distribution into groups, animals without clinical signs of health abnormalities were taken from cages, weighed, and marked with a temporary number. Body weight was used as a covariate. To achieve covariate balance across treatment groups, we sorted animals with temporary numbers by body weight in MS Excel. Animals were assigned to groups based on body weight stratification using the Latin square (2 × 2) method, and animals were assigned new numbers in accordance with the assigned group.

The animals from the first group were euthanized through inhaled CO_2_ and the animals from the second group received TZX intramuscularly into the quadriceps muscle of the thigh before blood sampling. The CO_2_ inhalation was performed according to the requirements for the euthanasia of laboratory rodents in a CO_2_ chamber [9,10]. All experiments were carried out at the beginning of 2020 in accordance with the AVMA 2013 recommendations [10]. The gas was fed to the chamber without a pre-filling at the rate of 20% of the volume of the chamber per minute, where the animals were kept for 3 min. The animals from the second group were anesthetized with Zoletil^®^ 100 (Virbac, Milan, Italy) in the combination with Rometar^®^ (2% xylazine, Bioveta, a.s., Ivanovicena Hané, Czeck Republic). Such a combination of Zoletil^®^ 100 and Rometar^®^ was approved by the Institutional Animal Care and Use Committee of BIBC RAS. The dose of the components was 12.5 mg/kg (for tiletamine and zolazepam) and 7.5 mg/kg (for xylazine), and the volume for intramuscular injection was adjusted based on the body weight of the animal so that the dose was the same (injection was about 0.05 mL per mice and 0.1 mL per rat).

The surgical stage of anesthesia 5–6 min after the injection was registered by the loss of the righting reflex, pedal reflex, and tail withdrawal after pinching. The loss of consciousness and reflexes of the animals in the CO_2_ chamber was ascertained after 5–6 min by the absence of respiratory activity through the transparent lid of the chamber. Immediately after the onset of the surgical stage of anesthesia after TZX injection or loss of animals’ respiratory activity in the CO_2_ chamber, the blood was sampled by laparotomy with a syringe from the vena cava inferior in the maximum possible volume, and was placed into a tube without an anticoagulant agent. The tubes were left at room temperature for 45–50 min for coagulation and centrifuged (1600× *g*, 4 °C, 15 min). The samples of serum were frozen (−20 °C) before the assay.

Automatic biochemical analyzer SAPPHIRE 400 (Tokyo Boeki LTD/Hirose Electronic Systems Co., Tokyo, Japan) with the reagents Randox Laboratories Ltd. (Crumlin, UK) was used for the assay of the following serum parameters: urea (the kinetic method with urease), creatinine (the method with alkaline picrate without deproteinization), cholesterol (the cholesterol oxidase method), triglycerides (the lipase and glycerokinase method), total protein (the biuret reaction method), albumin (bromocresol green method), globulins (calculated as a difference between the values of total protein and albumin), total bilirubin (the method with diazotized sulfanilate acid and caffeine), ALP (the IFCC-optimized p-nitrophenyl phosphate method), ALT and AST (the Tris buffer method without pyridoxal-5-phosphate, 37 °C), calcium (the arsenazo method), inorganic phosphates (the UV phosphomolybdate measurement method). Sodium, potassium and chloride ions were measured with an ion-selective block of the analyzer SAPPHIRE 400 and the respective electrodes for SAPPHIRE Series ISE. For the in-house routine quality control, commercial human reference serum was used (Randox Laboratories Ltd., Crumlin, United Kingdom). The laboratory takes part in the program of interlaboratory quality control of clinical biochemical changes (RIQAS: Randox International Quality Assessment Scheme). The normality of distribution was checked by the Shapiro–Wilk test. The cases with normal distribution were analyzed by a non-paired Student’s t-test separately for male and female animals, the rest of the cases were analyzed with a Mann–Whitney test. Animals of the same sex and the same species were compared in terms of one biochemical parameter, and the influence of one fixed factor×—CO_2_ or TZX treatment—on each biochemical parameter was studied. The effect of gender and animal species on biochemical parameters was not evaluated. Differences between groups were considered statistically significant if the *p*-value was <0.05.

To measure the level of glucose in the whole blood, the authors sampled a drop of blood from the tip of the tail of the awake animals. Sampling was repeated when animals were anesthetized by TZX or CO_2_ (right before terminal sampling). The level of glucose in the whole blood was tested with a glucometer OneTouch Select^®^ (LifeScan, Johnson & Johnson, Malvern, PA, USA). The significance of obtained differences was tested in two-way ANOVA, separately for one sex and one species (female mice, male mice, female rats, male rats) for which glucose levels were compared depending on the following fixed factors: (1) anesthesia with TZX/CO_2_ inhalation anesthesia; (2) awake animal/animal after terminal anesthesia. Confidence level at alpha 5% was determined by two-way ANOVA accompanied by a Duncan’s Multiple Range post-hoc test.

The data are presented as mean arithmetical and standard deviation (mean ± SD). The statistical analysis was performed with the program GraphPad Prism 6 (GraphPad Inc., San Diego, CA, USA). For statistically significant differences, we indicated (*) if *p* < 0.05, (**)—*p* < 0.01, (***)—*p* < 0.001.

## 3. Results

The surgical stage of anesthesia after the TZX administration was achieved averagely after 5 min (the minimum time was 4 min; the maximum was 7 min) both for rats and mice. The same average time (5–6 min) was necessary to detect the absence of respiratory activity of animals in the CO_2_ chamber (the heartbeat of the animals remained). Both rats and mice did not show distress or painful reactions during TZX administration or placement into the CO_2_ chamber, apnea and abnormal muscle activity were not observed either. Nevertheless, mice (Table 1) and rats’ (Table 2) blood samples showed a difference in analyzed biochemical parameters between the CO_2_ and TZX-treated groups.

The levels of total protein, albumin and globulins were significantly lower in male and female animals after the anesthesia with TZX in comparison with the inhalation of CO_2_. The ratio albumin:globulins in rats remained regardless of the method of anesthesia, and in mice—it was slightly higher by TZX (significantly in males). The level of urea was similar in rats and mice anesthetized by different methods. In rats of both sexes that received TZX, the level of creatinine was significantly lower. In mice, the differences in the level of creatinine between TZX and CO_2_-treated groups were not so expressive, but were significant for female mice. Similarly, the level of cholesterol and triglycerides was significantly lower in male and female rats, and only in female mice (the levels of cholesterol alone) for TXZ in comparison to the CO_2_ group. The measured activity of transaminases AST and ALT was decreased by animals’ anesthesia with TZX, and the level of ALP was lower only in the female rats’ group with a tiny tendency in male rats and mice of both sexes. The total bilirubin amount was significantly lower in groups that received TZX vs. the CO_2_ inhalation groups (excluding female rats).

The serums’ ions analysis showed the tendency to increase sodium and to decrease calcium, phosphates and chlorides ions for TZX group vs. CO_2_ group that was true for all animals of both sexes. The plausible difference for all ion content was measured for mice only, but the sharp decrease in potassium level was found between groups of all animals (3-fold drop in mice and female rats, 2.4-fold in male rats). As a result, the difference in samples was more noticeable for female mice than for male mice, and then for both rats.

Both CO_2_ inhalation and TZX caused approximately similar increases in glucose content in the blood of rats and mice of both sexes (Figure 1) from the level for awake animals. Only in male rats was hyperglycemia significantly expressed for TZX anesthesia over CO_2_ inhalation.

## 4. Discussion

The method of anesthesia and euthanasia of an animal during experiments can significantly influence the metabolism and physiological response of tissues, firstly, due to animal stress from experimental procedures, secondly due to the pharmacologic effect of the used agents and, finally, due to pre- and post-mortem metabolic changes. It is known that the increase in the level of glucose in blood is observed as a response to the expression of corticosterone under stress conditions, and is registered in laboratory rats during experimental procedures [29]. In the present study, the increase in glucose level was observed in rats and mice as a response to both procedures: CO_2_ inhalation and TZX injection. Acute hyperglycemic effect of CO_2_ inhalation can be associated with the sympathoadrenal activation and increased secretion of corticosterone. Earlier, it was proven that the level of corticosterone in rats’ plasma significantly increases when exposed to CO_2_ [15,17,30]. Restraining of an animal and the injection of anesthetic can also lead to stress and an increase in glucose levels in blood. At the same time, hyperglycemia can be induced by the pharmacologic effect of components. It is known that ketamine exerts a hyperglycemic effect associated with its sympathostimulating activity and activation of glycogenolysis [20,21,31], which can be enhanced in combination with α2-agonists. It was shown that the increase in the level of glucose in blood in rats and mice as a reaction to xylazine injection was associated with the decreased level of insulin caused by the activation of α2-receptors [21,32]. The mice that received ketamine–xylazine anesthesia had the level of glucose significantly increased in comparison with the mice that received CO_2_ inhalation [18]. The present study showed that the level of glucose in mice was similar after the CO_2_ inhalation and after TZX injection, and only in male rats anesthetized with TZX was hyperglycemia slightly more expressed (Appendix A). Tiletamine is used in combination only with zolazepam that exerts an anticonvulsive, anxiolytic and sedative effect, which inhibits the activation of the sympathetic system by tiletamine [27]. It is suggested that, in the present study, the level of glucose in mice and rats after TZX was associated primarily with the effect of xylazine. In male rats, the pharmacologic effect of xylazine can be more expressed.

After CO_2_ inhalation, a pronounced metabolic vasodilation develops within several minutes, which leads to the decrease in the perfusion of tissues and their hypoxia [33,34]. The inhibition of the Na+/K+ pump, induced by hypoxia, leads to the outflow of intercellular potassium and the passive transport of sodium into the cell [35]. The results of the present study agreed with data published by Traslavina et al. on female mice C57BL/6 after CO_2_ inhalation that showed pronounced acidosis, increase in the production of lactates, and significant hyperkaliemia in comparison with the mice that received anesthesia with ketamine–xylazine [13]. The level of potassium in rats and mice after TZX anesthesia agreed with the respective values obtained in other studies of rodents’ anesthesia without CO_2_ [7,36]. The inhalation of CO_2_ leads to significant hyperkaliemia and natriemia in male and female rats in comparison with the TZX anesthesia, which can be associated with the failure of the Na^+^/K^+^ pump caused by hypoxia.

Hyperkalemia in rats predisposes to both hyperexcitability of the heart (ventricular tachycardia, ventricular fibrillation) and depression (bradycardia, atrioventricular block, interventricular conduction delay, and asystole) [37]. However, it was reported [38] that prolonged induced hyperkalemia up to 10 mmol/L (with a normal potassium level of 5.3 mmol/L) in SHR rats does not lead to clinical distress.

CO_2_ inhalation in the chamber is a more stressful method and causes a cascade of pathophysiologic reactions reflected in blood, as well as the unnecessary suffering of animals [39,40]. In the condition of hypoxia after CO_2_ inhalation, hypercalcemia can be developed, as a consequence of electrolyte balance failure [41]. Besides, hypoxia can lead to a phosphate increase in serum due to the inhibition of the phosphateuretic effect of the parathyroid hormone [42], and a reduction in renal glomeruli filtration capacity.

The inhalation of CO_2_ led to an increase in the level of total protein in the serum of the group of rats and mice, which may be associated with animal stress. [43]. At the same time, after TZX injection, the level of protein could potentially decrease due to the possible reduction of cardiac output and the compensatory transport of fluid to the blood stream. In previous studies, TZX caused bradycardia and hypotension followed by a short-term increase in blood pressure associated with the central sympathoinhibiting effect of xylazine [44,45,46].

It can be suggested that the high levels of cholesterol and triglycerides after CO_2_ inhalation can be caused by the hemoconcentration effect, but can also be associated with the expression of the deposited blood from the liver and spleen, which is observed during the sympathetic activation and hypoxia [47,48]. Earlier, we showed that BALB/c mice had significantly higher amount of leukocyte, platelet, erythrocyte, and hematocrit at the same levels of MCH and MCHC [49] after the inhalation of CO_2_, in comparison with the injection of TZX.

The levels of transaminases ALT and AST are traditionally used as indicators of hepatocellular failure in rodents in toxicological studies [50,51]. However, the level of ALT in serum increases not only after the toxic damage of liver but also as a result of the general metabolic adaptive response to such effects that stimulate gluconeogenesis, such as stress or muscular activity [52,53]. AST is an enzyme that is not specific to the liver. It is contained in the majority of tissues, including erythrocytes. It should be noted that during the serum sampling, the authors registered the degree of hemolysis by the standard scale (0–4), and the increase in the degree of hemolysis was observed only in male mice after the inhalation of CO_2_ (median, 1.5 in comparison with 0 in the group TZX). Probably, higher values of AST and ALT in rodents after the inhalation of CO_2_ could be caused by changes in the metabolism, primarily, in the liver, as a response to acute hypoxia and stress. Earlier, the increase in the concentration of transaminases after the injection of ketamine–xylazine was registered in laboratory rats. However, if there were no histopathologic changes in the liver, these changes were more likely associated with the damage of muscles during injection, and a systemic effect of the combination of anesthetics [8,54].

There are three isoforms of ALP detected in rodents’ blood. In young mature Sprague Dawley rats, primarily an osteal and small intestine isoform is detected [55]; in the plasma of outbred mice, the main isoenzyme ALP is of osteal origin, but liver isoenzyme is also detected [56]. Still, the total level of ALP in rodents’ blood increases in the modeled conditions of cholestasis and directly correlates with the increase in the concentration of bile acids [57]. Higher levels of total bilirubin in rats and mice after the inhalation of CO_2_ in comparison with the injection of TZX can result from cholestasis. However, a significantly lower level of ALP in the group that received TZX was registered only in female rats and a similar tendency was observed in female mice. In male rats and mice, there were no significant differences observed after the application of different methods of anesthesia, probably, due to the sex-related dimorphism of the enzyme’s isoforms.

The buffer system of phosphocreatine–creatine is important for the maintenance of the level of ATP during hypoxia and the quick contraction of muscles during animal restraint. The level of creatinine increased in the skeletal muscles of euthanized mice [58]. Creatinine, detected in serum, is converted by means of the non-enzymatic transformation of creatine and can increase during hypoxia or muscle tissue damage during an injection [8], and can be an indicator of renal filtration failure or renal blood flow reduction. In the present study in rats, the level of creatinine was significantly higher after the inhalation of CO_2_ than after the TZX injection, probably, due to hypoxia and the reduction in the blood stream. In mice, the differences in the level of creatinine between the groups were not so expressed, which can be associated with the major damage of muscles during the injection and more severe failure of the renal filtration after anesthesia due to the small size of mice.

## 5. Conclusions

In the present study, we showed that the clinical biochemical parameters in the serum of outbred mice and rats significantly depended on the used method of terminal anesthesia. The serum content of total protein and albumin, cholesterol, triglycerides, aspartate aminotransferase (AST), alanine aminotransferase (ALT), alkaline phosphatase (ALP), total bilirubin, and creatinine was decreased by the injection of TZX in comparison with CO_2_ inhalation. In addition, the level of calcium, phosphates, chlorides, and potassium was lowered by TZX vs. CO_2_ administration, while the level of sodium increased. Preliminary anesthesia for small rodents with TZX for the clinical blood parameters assay is the most feasible method for achieving minimal influence on final data. The use of this method contributes to more reliable data obtaining for assessing both the toxic and therapeutic effect of drugs.

## Figures and Tables

**Figure 1 biomedicines-10-00512-f001:**
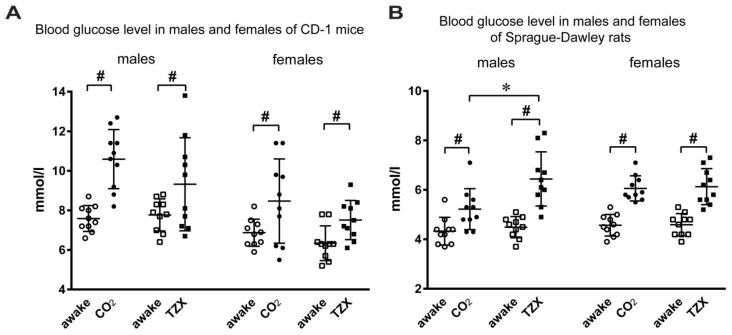
Blood glucose levels in males and females of CD-1 mice (**A**), and Sprague Dawley rats (**B**), in the awake state and following 5 min after CO_2_ inhalation or tiletamine–zolazepam–xylazine (TZX) anesthesia. # *p* < 0.05 TZX versus awake, * *p* < 0.05 TZX versus CO_2_.

**Table 1 biomedicines-10-00512-t001:** Clinical biochemistry results for serum of CD-1 mice collected in 5 min after CO_2_ inhalation or tiletamine–zolazepam–xylazine injection.

	Males		Females	
	CO_2_ (*n* = 10)	TZX (*n* = 10)	CO_2_ (*n* = 10)	TZX (*n* = 10)
Total protein (g/L)	54.5 ± 0.8	42.4 ± 0.8 ***	56.3 ± 1.1	42.3 ± 1.3 ***
Albumin (g/L)	33.5 ± 0.5	26.6 ± 0.4 ***	40.0 ± 1.8	31.7 ± 1.9 **
Globulins (g/L)	21.1 ± 0.4	15.8 ± 0.5 ***	16.2 ± 1.1	10.6 ± 0.8 **
Albumin:Globulin Ratio	1.6 ± 0.1	1.7 ± 0.1 *	2.7 ± 0.5	3.3 ± 0.4
Urea (mmol/L)	9.5 ± 0.4	9.1 ± 0.4	7.8 ± 0.2	9.0 ± 0.5
Creatinine (µmol/L)	38.7 ± 0.8	41.3 ± 1.1	48.7 ± 2.5	42.1 ± 1.8 *
Cholesterol (mmol/L)	3.29 ± 0.20	2.72 ± 0.19	2.73 ± 0.25	1.93 ± 0.11 *
Triglycerides (mmol/L)	1.50 ± 0.13	1.21 ± 0.08	2.16 ± 0.43	1.18 ± 0.12
Total bilirubin (µmol/L)	7.55 ± 0.73	5.50 ± 0.43 *	6.94 ± 0.82	6.07 ± 0.32 *
ALT (U/L)	58.9 ± 7.2	44.3 ± 6.4 *	44.1 ± 4.5	30.6 ± 1.6 *
AST (U/L)	77.6 ± 5.3	49.2 ± 3.0 ***	66.5 ± 2.5	53.3 ± 2.0 **
ALP (U/L)	76.9 ± 9.1	70.0 ± 8.8	143.6 ± 9.3	128.3 ± 11.7
Calcium (mmol/L)	2.81 ± 0.03	2.30 ± 0.02 ***	3.15 ± 0.09	2.44 ± 0.07 ***
Inorganic phosphorus (mmol/L)	3.87 ± 0.08	3.04 ± 0.13 ***	4.14 ± 0.25	3.23 ± 0.18 **
Sodium (mmol/L)	142.3 ± 1.6	152.2 ± 0.6 ***	144.5 ± 0.9	150.6 ± 0.6 ***
Potassium (mmol/L)	16.2 ± 0.4	5.2 ± 0.1 ***	13.5 ± 0.3	4.4 ± 0.1 ***
Chloride (mmol/L)	112.0 ± 1.3	110.4 ± 0.6 *	112.6 ± 0.6	110.2 ± 0.4 **

Data are presented as the mean ± SD; (*)—*p* < 0.05 TZX vs. CO_2_; (**)- *p* < 0.01 TZX vs. CO_2_; (***)—*p* < 0.001 TZX vs. CO_2_.

**Table 2 biomedicines-10-00512-t002:** Clinical biochemistry results for serum of Sprague Dawley rats collected in 5 min after CO_2_ inhalation or tiletamine–zolazepam–xylazine injection.

	Males		Females	
	CO_2_ (*n* = 10)	TZX (*n* = 10)	CO_2_ (*n* = 10)	TZX (*n* = 10)
Total protein (g/L)	70.9 ± 5.1	59.2 ± 4.6 ***	67.0 ± 3.7	57.6 ± 4.0 ***
Albumin (g/L)	41.0 ± 1.5	34.8 ± 2.2 ***	39.8 ± 1.5	35.1 ± 1.9 ***
Globulins (g/L)	29.8 ± 3.6	24.3 ± 3.0 **	27.1 ± 2.4	22.5 ± 2.4 ***
Albumin:Globulin Ratio	1.4 ± 0.1	1.4 ± 0.2	1.5 ± 0.1	1.6 ± 0.1
Urea (mmol/L)	9.5 ± 1.1	9.0 ± 1.5	7.9 ± 0.9	7.5 ± 1.1
Creatinine (µmol/L)	72.3 ± 4.5	52.2 ± 3.3 ***	70.0 ± 3.4	59.2 ± 4.3 ***
Cholesterol (mmol/L)	2.79 ± 0.45	1.98 ± 0.45 **	2.78 ± 0.38	2.23 ± 0.31 **
Triglycerides (mmol/L)	1.04 ± 0.26	0.62 ± 0.12 ***	0.94 ± 0.19	0.57 ± 0.12 ***
Total bilirubin (µmol/L)	5.23 ± 1.29	3.33 ± 0.80 **	3.93 ± 1.11	3.21 ± 0.82
ALT (U/L)	68.2 ± 10.1	52.1 ± 8.0 **	60.1 ± 11.9	37.8 ± 7.3 ***
AST (U/L)	83.8 ± 6.7	72.4 ± 8.3 *	83.7 ± 8.5	74.7 ± 6.7 **
ALP (U/L)	160.0 ± 27.0	138.1 ± 26.0	100.1 ± 26.3	72.3 ± 25.3 *
Calcium (mmol/L)	3.28 ± 0.27	2.51 ± 0.20 ***	2.94 ± 0.31	2.56 ± 0.15 **
Inorganic phosphorus (mmol/L)	4.41 ± 0.68	2.17 ± 0.33 ***	3.65 ± 0.92	1.75 ± 0.27 ***
Sodium (mmol/L)	141.9 ± 7.2	144.1 ± 6.0	132.0 ± 3.3	138.1 ± 2.3 ***
Potassium (mmol/L)	11.0 ± 1.5	4.6 ± 0.3 ***	12.5 ± 1.8	4.2 ± 0.3 ***
Chloride (mmol/L)	119.9 ± 7.8	117.2 ± 5.5	114.1 ± 2.1	112.8 ± 2.0

Data are presented as the mean ± SD; (*)—*p* < 0.05 TZX versus CO_2_; (**)—*p* < 0.01 TZX versus CO_2_; (***)—*p* < 0.001 TZX versus CO_2_.

## Data Availability

The data presented in this study are available in Appendix A.

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
