# Peer review of "Comparative Study of the Aftereffect of CO2 Inhalation or Tiletamine–Zolazepam–Xylazine Anesthesia on Laboratory Outbred Rats and Mice"

_biomedicines, 2022, doi:10.3390/biomedicines10020512_

Round 1
Reviewer 1 Report
In this study, authors aim to compare the stability of biochemical parameters between blood collection following terminal anaesthesia with CO2 vs TZX. The reason for this study should be reported in the abstract, but one has to go through the introduction to understand the problem being addressed.
There are a number issues to be addressed, before this manuscript can be deemed acceptable:
- There is no justification provided for the use of outbred mouse and rat strains, which should be duly justified, since these are more prone to higher variability in outcomes, without a gain in generalisation (see Festing 2010 DOI: 10.1177/0192623310373776, or Festing 2014:10.1093/ilar/ilu036).
- It is not clear why authors chose an injectable anaesthetic (instead of, for example, using a volatile agent, like Isoflurane), if part of the objective is to improve animal welfare, especially considering they used a volume that is on the upper end of the recommendations. The authors themselves recognise that restraining and injection can both affect welfare and alter the parameters being measured peri-mortem.
- No justification for the number of animals used is provided. No power calculation is provided.
- I assume the fixed factors in the reported 2-way ANOVA were method vs. sex, but this is not stated anywhere. Furthermore, the authors also make comparisons between species and "awake" vs "anaesthetised", and it should be made clear how these were fit into the statistical model.
- It makes no sense to report three levels for the α threshold. A decision has to be made before any study on what is an acceptable false-positive rate. Of course, lower p values give more confidence on the statistical significance of the differences, but one α only is to be provided. Moreover, this value has to be corrected for multiple comparisons, as detailed below.
- The authors compared 17 physiological parameters, between two drugs administered to 2 sexes of 2 species and between two time-points. This results in an enormous number of comparisons, and false "significant differences" are therefore bound to occur by random variation alone, and mistaken for true differences. Correction for multiple comparisons should thus be introduced to the statistical model, which may render some observed differences as non-significant. Making fewer comparisons (for instance, I would refrain from doing direct species comparisons and analyse them separately) is also an option.
- The fact that a fixed volume was administered to all animals is a possible source of bias. For example, females may be leaner than males of the same age, and therefore the dose/weight may be higher. It also adds experimental noise, since in the sample there might be high variability in weight. The authors report random allocation to groups (method of randomisation was not reported, and it should), which would even out said noise across groups, but in any case said noise is likely to be relevant and reduce statistical power, along other variable parameters from genetic variability of outbred strains.
- Figure 1 should have species and parameter being measured clearly indicated in the graphs, and not only in the legend.
- The English should be extensively revised. Examples include (non-exhaustive list):
- 20-"aim of this study was in estimation"
- 28-"over estimation", rather than "overestimation"
- 51-starting a paragraph with "besides"
- 55-"injection anesthesia", rather than "injectable anesthesia"
- 62-64-not even sure what it means (and reference to gazelles is moreover inadequate)
- 64-"bi", rather than large
- 73-"bloods parameters", rather than "blood parameters"
- 74- unclear what authors mean by "to find out possible use TZX"
- 94 remove "the" before "wood fibers"
- 96-remove "the" before "blood sampling"
- 101-idem
- 102-replace "to the" with "for"
- 116 remove "the" before laparatomy
- 118 remove "the" before "room temperature"
- 122 remove "the" before "sampling"
- 173 replace "expressed" with "expressive"
- 184 replace "dramatically" by "sharp"
- replace "3 times" and "2.4 times" with "3-fold" and 2.4-fold, and "fall" with "drop"
- 200, 210 and 312 - not sure "stressogenity" is an actual word
- 209 | 294 replace "fixation" with "restraining"
- 246 replace "increase in the phosphates" with "phosphates increase"
- 250 - not sure what authors mean, here
- replace "earlier" with "in previous studies"
- 258-remove "the" before CO2
- 261-replace "the authors" with "we have shown"
- 262- replace "mice Balb/c" with "Balb/c mice"
- 266-replace "but" with "however"
- 283 remove "the before "osteal"
- 294-remove "the" before "hypoxia"
- 311-314-phrase is unintelligible
Author Response
Reviewer 1, point to point respond.
In this study, authors aim to compare the stability of biochemical parameters between blood collection following terminal anaesthesia with CO2 vs TZX. The reason for this study should be reported in the abstract, but one has to go through the introduction to understand the problem being addressed.
The necessary clarification was made in the abstract of the article
There are a number issues to be addressed, before this manuscript can be deemed acceptable:
- There is no justification provided for the use of outbred mouse and rat strains, which should be duly justified, since these are more prone to higher variability in outcomes, without a gain in generalisation (see Festing 2010 DOI: 10.1177/0192623310373776, or Festing 2014:10.1093/ilar/ilu036).
Today, there is a certain international practice in the use of outbred animals in toxicological studies, which are more resistant to the toxicity of the studied drugs, and therefore give a more realistic data compared to linear animals. We decided to use outbred stock of animals as the most common case in practice , despite the risk of obtaining less consistent results. Explanation included in the manuscript (introduction).
- It is not clear why authors chose an injectable anaesthetic (instead of, for example, using a volatile agent, like Isoflurane), if part of the objective is to improve animal welfare, especially considering they used a volume that is on the upper end of the recommendations. The authors themselves recognise that restraining and injection can both affect welfare and alter the parameters being measured peri-mortem.
When using isoflurane inhalation, the animal recovers from anesthesia less predictably and faster (after about 2 minutes after inhalation is stopped), which makes this method less acceptable before necropsy of the animal, due to the possibility of awakening the animal before or at the time of blood sampling. In addition, the higher cost of using isoflurane and similar gas anesthetics has made their use less common in practice today. The corresponding phrase is added to the article.
- No justification for the number of animals used is provided. No power calculation is provided.
We used 10 animals in each group. This number of animals was sufficient for subsequent statistical analysis. To determine the size of the groups, we carried out calculations in the Sample Size / Power Calculations program at the link http://www.biomath.info/power/index.html, which is in the public domain. Based on the calculation of the variability of each biochemical parameter, we determined that the minimum required number of animals is 10. The corresponding explanation is included in the text of the article in Materials and Methods.
- I assume the fixed factors in the reported 2-way ANOVA were method vs. sex, but this is not stated anywhere. Furthermore, the authors also make comparisons between species and "awake" vs "anaesthetised", and it should be made clear how these were fit into the statistical model.
We did not analyze the effect of animals' sex on glucose levels, so we made an appropriate clarification in the description of the method - to determine the effect of the anesthesia method on blood glucose levels, we compared within four independent groups (female mice, male mice, female rats, male rats).
- It makes no sense to report three levels for the α threshold. A decision has to be made before any study on what is an acceptable false-positive rate. Of course, lower p values give more confidence on the statistical significance of the differences, but one α only is to be provided. Moreover, this value has to be corrected for multiple comparisons, as detailed below.
Thanks for the comments. Historically, we calculate significant differences in single comparision analyses for several P values. Unless the reviewer insists categorically, we would like to leave it as is.
- The authors compared 17 physiological parameters, between two drugs administered to 2 sexes of 2 species and between two time-points. This results in an enormous number of comparisons, and false "significant differences" are therefore bound to occur by random variation alone, and mistaken for true differences. Correction for multiple comparisons should thus be introduced to the statistical model, which may render some observed differences as non-significant. Making fewer comparisons (for instance, I would refrain from doing direct species comparisons and analyse them separately) is also an option.
Thank you for your comment, we always compared the values in tables 1 and 2 between CO2-and TZX groups (this is indicated in the caption), namely, 4 independent analyzes are presented in the tables.For ease perception, tables have been redesigned, and now compared columns are located side by side. The note about the time is not correct, we maintained the same sampling interval.
- The fact that a fixed volume was administered to all animals is a possible source of bias. For example, females may be leaner than males of the same age, and therefore the dose/weight may be higher. It also adds experimental noise, since in the sample there might be high variability in weight. The authors report random allocation to groups (method of randomisation was not reported, and it should), which would even out said noise across groups, but in any case said noise is likely to be relevant and reduce statistical power, along other variable parameters from genetic variability of outbred strains.
Clarifications regarding the method of randomization, which was carried out taking into account the body weight. were made in the text. We have also corrected the phrase regarding the volume of the injected sample -we always corrected the volume of administration for the weight of the animal, earlier in the article it was erroneously written that it was constant.
- Figure 1 should have species and parameter being measured clearly indicated in the graphs, and not only in the legend.
We indicated the measured parameter and animals in the figure 1.
- The English should be extensively revised. Examples include (non-exhaustive list):
- 20-"aim of this study was in estimation" - phrase has been corrected
- 28-"over estimation", rather than "overestimation" - is corrected
- 51-starting a paragraph with "besides" - phrase has been corrected
- 55-"injection anesthesia", rather than "injectable anesthesia" - is corrected
- 62-64-not even sure what it means (and reference to gazelles is moreover inadequate) - phrase has been corrected
- 64-"bi", rather than large - " - is corrected
- 73-"bloods parameters", rather than "blood parameters" " - is corrected
- 74- unclear what authors mean by "to find out possible use TZX" - phrase has been corrected
- 94 remove "the" before "wood fibers"" - is corrected
- 96-remove "the" before "blood sampling"" - is corrected
- 101-idem" - is corrected
- 102-replace "to the" with "for"" - is corrected
- 116 remove "the" before laparatomy " - is corrected
- 118 remove "the" before "room temperature" " - is corrected
- 122 remove "the" before "sampling"" - is corrected
- 173 replace "expressed" with "expressive" " - is corrected
- 184 replace "dramatically" by "sharp" " - is corrected
- replace "3 times" and "2.4 times" with "3-fold" and 2.4-fold, and "fall" with "drop" " - is corrected
- 200, 210 and 312 - not sure "stressogenity" is an actual word
- 209 | 294 replace "fixation" with "restraining" " - is corrected
- 246 replace "increase in the phosphates" with "phosphates increase" " - is corrected
- 250 - not sure what authors mean, here – phrase has been changed
- replace "earlier" with "in previous studies" " - is corrected
- 258-remove "the" before CO2 " - is corrected
- 261-replace "the authors" with "we have shown" " - is corrected
- 262- replace "mice Balb/c" with "Balb/c mice" " - is corrected
- 266-replace "but" with "however" " - is corrected
- 283 remove "thebefore "osteal" " - is corrected
- 294-remove "the" before "hypoxia"" - is corrected
- 311-314-phrase isunintelligible - conclusion is changed
Thank you very much for the careful reading of our work, all the comments made by us have been eliminated.
Reviewer 2 Report
This paper shows comparative data in serum contents in different anesthetic methods between CO2 inhalation and injectable anesthesia. Data may be helpful for researchers to investigate serum contents in rodents. However, the following issues should be improved before acceptance.
- Reference 5, AVMA Guideline for the Euthanasia of Animals, 2013 version was already revised in 2020. Authors should refer the latest version. Further, recommended rate of CO2 filling has been changed from 10-30% to 30-70%. However, rate of CO2 filling was used 20% in this paper. Authors should clarify this difference in the discussion.
- In conclusions, authors mention that CO2 inhalation causes unnecessary suffering of animals. However, it is not clear to conclude it from data in this paper only.
- In Line 83-84, authors spell out AAALAC International. However, AAALAC International abolished such spelling out in 2016.
Author Response
Reviewer 2, point to point respond.
This paper shows comparative data in serum contents in different anesthetic methods between CO2 inhalation and injectable anesthesia. Data may be helpful for researchers to investigate serum contents in rodents. However, the following issues should be improved before acceptance.
- Reference 5, AVMA Guideline for the Euthanasia of Animals, 2013 version was already revised in 2020. Authors should refer the latest version. Further, recommended rate of CO2 filling has been changed from 10-30% to 30-70%. However, rate of CO2 filling was used 20% in this paper. Authors should clarify this difference in the discussion.
Unfortunately, the global situation with the pandemic has made adjustments to our work, so now we are publishing the data received at the beginning of 2020. The experiment was carried out in accordance with AVMA 2013 recommendations, this fact is given in materials and methods.
- In conclusions, authors mention that CO2 inhalation causes unnecessary suffering of animals. However, it is not clear to conclude it from data in this paper only.
In the study, we did not record the suffering of animals during CO2 inhalation.We transferred this information from the conclusion to the discussions, where we provided additional references with such observations.
- In Line 83-84, authors spell out AAALAC International. However, AAALAC International abolished such spelling out in 2016
Spelling is corrected
Round 2
Reviewer 1 Report
The authors will find my comment on their reply to my original review in bold. In most cases, the authors have failed to address the main issues of the paper and are urged to not avoid said issues and do the necessary changes.
In this study, authors aim to compare the stability of biochemical parameters between blood collection following terminal anaesthesia with CO2 vs TZX. The reason for this study should be reported in the abstract, but one has to go through the introduction to understand the problem being addressed.
The necessary clarification was made in the abstract of the article
OK
There are a number issues to be addressed, before this manuscript can be deemed acceptable:
- There is no justification provided for the use of outbred mouse and rat strains, which should be duly justified, since these are more prone to higher variability in outcomes, without a gain in generalisation (see Festing 2010 DOI: 10.1177/0192623310373776, or Festing 2014:10.1093/ilar/ilu036).
Today, there is a certain international practice in the use of outbred animals in toxicological studies, which are more resistant to the toxicity of the studied drugs, and therefore give a more realistic data compared to linear animals. We decided to use outbred stock of animals as the most common case in practice , despite the risk of obtaining less consistent results. Explanation included in the manuscript (introduction).
No reference is provided in support of this added statement on the choice for outbred animals, and it should be.
- It is not clear why authors chose an injectable anaesthetic (instead of, for example, using a volatile agent, like Isoflurane), if part of the objective is to improve animal welfare, especially considering they used a volume that is on the upper end of the recommendations. The authors themselves recognise that restraining and injection can both affect welfare and alter the parameters being measured peri-mortem.
When using isoflurane inhalation, the animal recovers from anesthesia less predictably and faster (after about 2 minutes after inhalation is stopped), which makes this method less acceptable before necropsy of the animal, due to the possibility of awakening the animal before or at the time of blood sampling. In addition, the higher cost of using isoflurane and similar gas anesthetics has made their use less common in practice today. The corresponding phrase is added to the article.
OK
- No justification for the number of animals used is provided. No power calculation is provided.
We used 10 animals in each group. This number of animals was sufficient for subsequent statistical analysis. To determine the size of the groups, we carried out calculations in the Sample Size / Power Calculations program at the link http://www.biomath.info/power/index.html, which is in the public domain. Based on the calculation of the variability of each biochemical parameter, we determined that the minimum required number of animals is 10. The corresponding explanation is included in the text of the article in Materials and Methods.
The authors are cautioned to explain why 10 animals per groups were deemed the minimum necessary in their power calculation. What was the predefined alpha value set in the power calculation (I presume 0.05 but should be made clear), for what power (80%? 90% other?) for what tests, and for what effect sizes?
- I assume the fixed factors in the reported 2-way ANOVA were method vs. sex, but this is not stated anywhere. Furthermore, the authors also make comparisons between species and "awake" vs "anaesthetised", and it should be made clear how these were fit into the statistical model.
We did not analyze the effect of animals' sex on glucose levels, so we made an appropriate clarification in the description of the method - to determine the effect of the anesthesia method on blood glucose levels, we compared within four independent groups (female mice, male mice, female rats, male rats).
This description is confusing, both in the reply and the revised manuscript. If two-way ANOVA is used, two fixed factors and their interactions are tested for significance. Authors should state which were the fixed factors entered into the statistical model. In my view, it would make more sense to analyse species separately (i.e., as if for each species we have separate studies), and within each species to compare anaesthesia methods and check for sex differences and interactions with drug. Otherwise, why use this factorial arrangement in the first place?
I therefore urge authors to:
- State clearly what were the hypotheses being tested. (as this is what determines what are the fixed factors in the statistical model and the parameters being measured)
- Describe the experimental design in light of the hypotheses being tested
- Describe which factors are being tested, and why.
- Make sure that corrections for multiple comparisons are introduced to the statistical model, and describe what correction was applied
- It makes no sense to report three levels for the α A decision has to be made before any study on what is an acceptable false-positive rate. Of course, lower p values give more confidence on the statistical significance of the differences, but one α only is to be provided. Moreover, this value has to be corrected for multiple comparisons, as detailed below.
Thanks for the comments. Historically, we calculate significant differences in single comparision analyses for several P values. Unless the reviewer insists categorically, we would like to leave it as is.
I insist that it makes no sense to state that "The data was statistically different at P<0.05, P < 0.01 or P < 0.001". It makes no sense. It is as if authors are not even aware of what an alpha value is. The setting of ONE alpha value is done a priori and every p value below it is considered to be significant. If p value is lower than the selected threshold (e.g. α=0.05) it should be duly reported, but for the sake of the determining whether it is significant or not, the criteria is whether it is below the predefined alpha threshold.
- The authors compared 17 physiological parameters, between two drugs administered to 2 sexes of 2 species and between two time-points. This results in an enormous number of comparisons, and false "significant differences" are therefore bound to occur by random variation alone, and mistaken for true differences. Correction for multiple comparisons should thus be introduced to the statistical model, which may render some observed differences as non-significant. Making fewer comparisons (for instance, I would refrain from doing direct species comparisons and analyse them separately) is also an option.
Thank you for your comment, we always compared the values in tables 1 and 2 between CO2-and TZX groups (this is indicated in the caption), namely, 4 independent analyzes are presented in the tables.For ease perception, tables have been redesigned, and now compared columns are located side by side. The note about the time is not correct, we maintained the same sampling interval.
I find there is no justification for not applying corrections for multiple comparisons (e.g. Bonferroni, Šidák). Reanalyse the data doing so. It might lead to fewer "significant differences", but those found will be more reliable.
- The fact that a fixed volume was administered to all animals is a possible source of bias. For example, females may be leaner than males of the same age, and therefore the dose/weight may be higher. It also adds experimental noise, since in the sample there might be high variability in weight. The authors report random allocation to groups (method of randomisation was not reported, and it should), which would even out said noise across groups, but in any case said noise is likely to be relevant and reduce statistical power, along other variable parameters from genetic variability of outbred strains.
Clarifications regarding the method of randomization, which was carried out taking into account the body weight. were made in the text.
The explanation about randomization method is not satisfactory. Were animals just taken “randomly” out of the cage (this is not proper randomization)? Was a software used (e.g. RAND function in MS Excel)? Were tiny papers with the code for each animal taken out of a bag? It is unclear. Also, saying that body weight was taken into account says absolutely nothing. How was it taken into account? And did you introduced it as a covariate?
We have also corrected the phrase regarding the volume of the injected sample -we always corrected the volume of administration for the weight of the animal, earlier in the article it was erroneously written that it was constant.
Where is this stated? The correction the authors made was the exact opposite of what they are stating here, as they say in line 121 that they administered an “average volume”, rather than adjusted to weight.
Author Response
Reviewer 1, point to point respond, round 2.
The authors will find my comment on their reply to my original review in bold. In most cases, the authors have failed to address the main issues of the paper and are urged to not avoid said issues and do the necessary changes.
In this study, authors aim to compare the stability of biochemical parameters between blood collection following terminal anaesthesia with CO2 vs TZX. The reason for this study should be reported in the abstract, but one has to go through the introduction to understand the problem being addressed.
The necessary clarification was made in the abstract of the article
OK
There are a number issues to be addressed, before this manuscript can be deemed acceptable:
- There is no justification provided for the use of outbred mouse and rat strains, which should be duly justified, since these are more prone to higher variability in outcomes, without a gain in generalisation (see Festing 2010 DOI: 10.1177/0192623310373776, or Festing 2014:10.1093/ilar/ilu036).
Today, there is a certain international practice in the use of outbred animals in toxicological studies, which are more resistant to the toxicity of the studied drugs, and therefore give a more realistic data compared to linear animals. We decided to use outbred stock of animals as the most common case in practice , despite the risk of obtaining less consistent results. Explanation included in the manuscript (introduction).
No reference is provided in support of this added statement on the choice for outbred animals, and it should be.
Yes, Festing reports that there is a need to reconsider the use of outbred animals in toxicological research, but there hasn't been a significant turnaround so far and outbred stocks are being used as well. Outbred mice are popular in toxicology and specific activity studies along with inbred and genetic lines (https://doi.org/10.5625/lar.2017.33.1.8, doi:10.1371/journal.pone.0004729). Outbred rats are currently being used to study various types of toxicity (doi:10.1002/bdr2.1889, doi:10.1016/j.toxrep.2021.10.010, https://doi.org/10.1093/ilar/ilu036). We have added the necessary references and explanations to the text of the manuscript (Introduction).
- It is not clear why authors chose an injectable anaesthetic (instead of, for example, using a volatile agent, like Isoflurane), if part of the objective is to improve animal welfare, especially considering they used a volume that is on the upper end of the recommendations. The authors themselves recognise that restraining and injection can both affect welfare and alter the parameters being measured peri-mortem.
When using isoflurane inhalation, the animal recovers from anesthesia less predictably and faster (after about 2 minutes after inhalation is stopped), which makes this method less acceptable before necropsy of the animal, due to the possibility of awakening the animal before or at the time of blood sampling. In addition, the higher cost of using isoflurane and similar gas anesthetics has made their use less common in practice today. The corresponding phrase is added to the article.
OK
- No justification for the number of animals used is provided. No power calculation is provided.
We used 10 animals in each group. This number of animals was sufficient for subsequent statistical analysis. To determine the size of the groups, we carried out calculations in the Sample Size / Power Calculations program at the link http://www.biomath.info/power/index.html, which is in the public domain. Based on the calculation of the variability of each biochemical parameter, we determined that the minimum required number of animals is 10. The corresponding explanation is included in the text of the article in Materials and Methods.
The authors are cautioned to explain why 10 animals per groups were deemed the minimum necessary in their power calculation. What was the predefined alpha value set in the power calculation (I presume 0.05 but should be made clear), for what power (80%? 90% other?) for what tests, and for what effect sizes?
Some of the clinical pathology parameters have relatively high variance (ALP, cholesterol, triglycerides). Using Power Analysis, to ensure parameters 80% power and 5% alpha the minimum number of animals in groups was 10. Power Analysis was performed for each biochemical parameter separately within each group and is given in the table below, which, if necessary, we can add to additional materials.
|
Parameter |
Effect size (difference between means) |
Male rats SD |
Male rats Group size |
Female rats SD |
Female rats Group size |
Male mice SD |
Male mice Group size |
Female mice SD |
Female mice Group size |
|
Total protein (g/L) |
5 |
1.4 |
<6 |
3.5 |
9 |
3.0 |
7 |
3.2 |
8 |
|
Albumin (g/L) |
3 |
0.8 |
<6 |
1.7 |
7 |
1.0 |
<6 |
1.2 |
<6 |
|
Globulins (g/L) |
3 |
1.2 |
<6 |
1.9 |
8 |
1.5 |
<6 |
1.5 |
<6 |
|
Albumin:Globulin Ratio |
0.3 |
0.1 |
<6 |
0.1 |
<6 |
0.1 |
<6 |
0.1 |
<6 |
|
Urea (mmol/L) |
6 |
0.5 |
<6 |
1.3 |
<6 |
1.8 |
7 |
1.1 |
<6 |
|
Creatinine (µmol/L) |
10 |
3 |
<6 |
7 |
9 |
4 |
<6 |
6 |
7 |
|
Cholesterol (mmol/L) |
0.35 |
0.25 |
10 |
0.24 |
8 |
0.25 |
10 |
0.25 |
10 |
|
Triglycerides (mmol/L) |
0.2 |
0.15 |
10 |
0.15 |
10 |
0.11 |
6 |
0.13 |
8 |
|
Total bilirubin (µmol/L) |
2 |
1.2 |
7 |
0.6 |
<6 |
1.1 |
6 |
1.5 |
10 |
|
ALT (U/L) |
30 |
14 |
<6 |
14 |
<6 |
18 |
7 |
21 |
9 |
|
AST (U/L) |
30 |
19 |
8 |
22 |
10 |
15 |
<6 |
14 |
<6 |
|
ALP (U/L) |
80 |
57 |
9 |
29 |
<6 |
50 |
8 |
54 |
9 |
|
Calcium (mmol/L) |
0.2 |
0.07 |
<6 |
0.15 |
10 |
0.1 |
<6 |
0.1 |
<6 |
|
Inorganic phosphorus (mmol/L) |
0.5 |
0.35 |
9 |
0.24 |
<6 |
0.28 |
6 |
0.22 |
<6 |
|
Sodium (mmol/L) |
2.5 |
1.7 |
9 |
1.8 |
10 |
1.8 |
10 |
1.3 |
6 |
|
Potassium (mmol/L) |
1 |
0.67 |
9 |
0.74 |
10 |
0.75 |
10 |
0.6 |
6 |
|
Chloride (mmol/L) |
2 |
1.5 |
10 |
1.5 |
10 |
1.4 |
9 |
1.5 |
10 |
- I assume the fixed factors in the reported 2-way ANOVA were method vs. sex, but this is not stated anywhere. Furthermore, the authors also make comparisons between species and "awake" vs "anaesthetised", and it should be made clear how these were fit into the statistical model.
We did not analyze the effect of animals' sex on glucose levels, so we made an appropriate clarification in the description of the method - to determine the effect of the anesthesia method on blood glucose levels, we compared within four independent groups (female mice, male mice, female rats, male rats).
This description is confusing, both in the reply and the revised manuscript. If two-way ANOVA is used, two fixed factors and their interactions are tested for significance. Authors should state which were the fixed factors entered into the statistical model. In my view, it would make more sense to analyse species separately (i.e., as if for each species we have separate studies), and within each species to compare anaesthesia methods and check for sex differences and interactions with drug. Otherwise, why use this factorial arrangement in the first place?
I therefore urge authors to:
- State clearly what were the hypotheses being tested. (as this is what determines what are the fixed factors in the statistical model and the parameters being measured)
- Describe the experimental design in light of the hypotheses being tested
- Describe which factors are being tested, and why.
- Make sure that corrections for multiple comparisons are introduced to the statistical model, and describe what correction was applied
- We applied different hypotheses for the glucose level and for the main biochemical parameters.
Parameter - glucose level:
We compared the effects of TZX anesthesia and CO2 inhalation on glucose values in awake animals and those in the anesthetized state. Two-way ANOVA was used. Factors that influenced glucose levels were: (1) TZX anesthesia/CO2 inhalation, (2) awake animal/anesthetized animal. The influence of these factors was compared separately for female mice, male mice, female rats, and male rats. It was possible to compare the anesthetized and awake animal, since glucometry uses minimal blood volumes, for blood biochemical parameters this was not feasible.
Biochemical parameters
We tested the hypothesis of how TZX anesthesia or CO2 inhalation would affect each biochemical parameter separately (factor - type of anesthesia - using TZX or CO2). Before making a comparison for each indicator separately, we checked the normality of the distribution in the Shapiro-Wilk test. Within one species of the same sex, the effect of CO2 and TZX was compared separately for each parameter of clinical biochemistry in the Student's t-test or Mann-Whitney test, if the distribution was not normal (i.e., the cholesterol level in female rats that were anesthetized with TZX was compared with the cholesterol level in female rats that have been exposed to CO2 inhalation, etc.).
- For glucose levels: Separately, glucose levels were studied within one species and one sex. Glucose levels were measured in the waking animals. Then, half of the animals were anesthetized with TZX or CO2, and glucose levels were measured again. Using two-way ANOVA, separately for one sex and one species (female mice, male mice, female rats, male rats), glucose levels were compared depending on wakefulness/anesthesia, as well as depending on exposure to TZX or CO2.
Other biochemical parameters were also separately studied in female mice, male mice, female rats, and male rats. Half of the animals received TZX mixture, the other half of the animals received CO2 inhalation. For each biochemical parameter, the normality of distribution within one species and sex was checked. In the Student's t-test or Mann-Whitney test, animals of the same sex, of the same species, but with different types of anesthesia, CO2 or TZX, were compared separately for one biochemical parameter.
- For glucose levels, differences in the awake/anesthetized animal and TZX/CO2 exposure factor were tested. Primary data that were obtained from glucomertia are presented in supplementary materials (Table S1, Table S2), two-way ANOVA was used to analyze these data.
For other biochemical parameters, only the TZX/CO2 exposure factor was tested, so a one-way analysis of the reliability of the results was performed. The level of 95% (P < 0.05) was taken as a significant difference.
- Duncan's post-hoc test was performed for two-way ANOVA when comparing glucose levels.
These clarifications were added to the materials and methods in a less detailed form and the paragraphs were rearranged.
- It makes no sense to report three levels for the α A decision has to be made before any study on what is an acceptable false-positive rate. Of course, lower p values give more confidence on the statistical significance of the differences, but one α only is to be provided. Moreover, this value has to be corrected for multiple comparisons, as detailed below.
Thanks for the comments. Historically, we calculate significant differences in single comparision analyses for several P values. Unless the reviewer insists categorically, we would like to leave it as is.
I insist that it makes no sense to state that "The data was statistically different at P<0.05, P < 0.01 or P < 0.001". It makes no sense. It is as if authors are not even aware of what an alpha value is. The setting of ONE alpha value is done a priori and every p value below it is considered to be significant. If p value is lower than the selected threshold (e.g. α=0.05) it should be duly reported, but for the sake of the determining whether it is significant or not, the criteria is whether it is below the predefined alpha threshold.
One level of significance is left in the results.
- The authors compared 17 physiological parameters, between two drugs administered to 2 sexes of 2 species and between two time-points. This results in an enormous number of comparisons, and false "significant differences" are therefore bound to occur by random variation alone, and mistaken for true differences. Correction for multiple comparisons should thus be introduced to the statistical model, which may render some observed differences as non-significant. Making fewer comparisons (for instance, I would refrain from doing direct species comparisons and analyse them separately) is also an option.
Thank you for your comment, we always compared the values in tables 1 and 2 between CO2-and TZX groups (this is indicated in the caption), namely, 4 independent analyzes are presented in the tables. For ease perception, tables have been redesigned, and now compared columns are located side by side. The note about the time is not correct, we maintained the same sampling interval.
I find there is no justification for not applying corrections for multiple comparisons (e.g. Bonferroni, Љidбk). Reanalyse the data doing so. It might lead to fewer "significant differences", but those found will be more reliable.
For biochemical parameters We did not aim to reveal the dependence on the sex and species of animals, therefore, we used a one-parameter comparison in the Student's t-test in animals of the same species and the same sex. Multiple comparisons between species or sexes were not made due to significant fluctuations in the mean values of the parameters between these groups.
Duncan's post-hoc test was performed to analyze glucose concentration in two-way ANOVA. The previous difficulties in interpreting the methodology have been clarified in the new version of the manuscript.
- The fact that a fixed volume was administered to all animals is a possible source of bias. For example, females may be leaner than males of the same age, and therefore the dose/weight may be higher. It also adds experimental noise, since in the sample there might be high variability in weight. The authors report random allocation to groups (method of randomisation was not reported, and it should), which would even out said noise across groups, but in any case said noise is likely to be relevant and reduce statistical power, along other variable parameters from genetic variability of outbred strains.
Clarifications regarding the method of randomization, which was carried out taking into account the body weight. were made in the text.
The explanation about randomization method is not satisfactory. Were animals just taken “randomly” out of the cage (this is not proper randomization)? Was a software used (e.g. RAND function in MS Excel)? Were tiny papers with the code for each animal taken out of a bag? It is unclear. Also, saying that body weight was taken into account says absolutely nothing. How was it taken into account? And did you introduced it as a covariate?
Animals were randomly taken from cages, weighed, and assigned a temporary number. Body weight data with animal temporary numbers were entered into MS Excel. Body weight separately for each species and each sex was sorted in ascending order. Further, using the block design method, the animal was enrolled in a group and assigned a number in the study. Thus, we had a list of body weights sorted in MS Excel in ascending order, and each animal, in the order of this list, was assigned to the following groups: 1, 2; 2, 1, etc. The explanation is included in the text of the article. We have also corrected the phrase regarding the volume of the injected sample -we always corrected the volume of administration for the weight of the animal, earlier in the article it was erroneously written that it was constant.
Where is this stated? The correction the authors made was the exact opposite of what they are stating here, as they say in line 121 that they administered an “average volume”, rather than adjusted to weight.
Thank you, we corrected the error in the text.

Round 3
Reviewer 1 Report
Dear authors,
Thank you for reviewing the manuscript.
I have the following final comments, to be addressed before deeming this manuscript publishable:
line 39 - Please replace "purebred" for "inbred" or, better yet, "isogenic".
line 108 - you mentioned weight categories as blocks, but not describe how many blocks you divided animals into, or whether these were added as random factors in the analysis (also, and in my opinion, weight would be best used as a covariate, rather than categorized and turned into a block). Please do so.
Line 109 - Just a comment, as you cannot re-do your study. I recommend that for future studies randomization is done prior to assigning animals to groups (i.e. assign each animal a number/reference, randomize treatments using appropriate software and only then assign animals to the treatments). Picking animals at "random" is not proper randomization because there might be a reason for why some animals allow themselves to be caught first and/or unblinded observers have a subconscious order of preference, and this might bias results, as it may become an undetected "grouping variable".
Line 153. I find it strange that you would go through the trouble of using both sexes (which is commendable), but then not use it as a factor in the analysis to check whether there is a sex-dependent effect, raising the issue of why use two sexes in the first place. I will not again insist on it, but I do recommend you reconsider using sex as a factor, as it gives more information without need for more animals. If not for this study, for future ones.
I think you misunderstood what I meant in my previous review regarding your reporting of more than one alpha value for significance in your methods section. One thing is the threshold for significance (alpha), which must be referred to in the methods, the other is the p value you obtain as a test statistic, which should be reported as it is (fir example p<0.05, p<0.01, p<0.001, p=0.034, etc.). You must only have one alpha (e.g. 0.05, 0.005 or 0.001) to determine what you consider to be significant, but you can report your actual p values, so there is no need to change them in the results, just the predefined alpha in your methods section (so please revert these to what you had previously reported, in the results section). Again, authors are advised to read on the meaning of alpha and p values. I suggest the following blog post by D. Lakens: http://daniellakens.blogspot.com/2021/11/why-p-values-should-be-interpreted-as-p.html
Author Response
Reviewer 1, point to point respond, round 3.
Thank you for reviewing the manuscript.
I have the following final comments, to be addressed before deeming this manuscript publishable:
line 39 - Please replace "purebred" for "inbred" or, better yet, "isogenic". – is corrected
line 108 - you mentioned weight categories as blocks, but not describe how many blocks you divided animals into, or whether these were added as random factors in the analysis (also, and in my opinion, weight would be best used as a covariate, rather than categorized and turned into a block). Please do so.
In this work, we used body weight as a covariate. To achieve covariate balance across treatment groups, we sorted the list of body weights in ascending order, and distributed each animal to the order of the sorted list using the Latin square (2x2) method. That is, the first animal with the minimum weight was placed in group 1, the second animal from the list with the smallest weight was placed in group 2, the next in group 2, and the next in group 1. We did not distribute animals into blocks according to body weight, the wording in the manuscript was corrected.
Line 109 - Just a comment, as you cannot re-do your study. I recommend that for future studies randomization is done prior to assigning animals to groups (i.e. assign each animal a number/reference, randomize treatments using appropriate software and only then assign animals to the treatments). Picking animals at "random" is not proper randomization because there might be a reason for why some animals allow themselves to be caught first and/or unblinded observers have a subconscious order of preference, and this might bias results, as it may become an undetected "grouping variable". –
After the stratification of animals with temporary numbers by body weight and distribution according to the Latin square, the animal was assigned a group and a new final number, which was included in the study. In the text of the manuscript, the “random” was removed to more correctly reflect the methodology.
Line 153. I find it strange that you would go through the trouble of using both sexes (which is commendable), but then not use it as a factor in the analysis to check whether there is a sex-dependent effect, raising the issue of why use two sexes in the first place. I will not again insist on it, but I do recommend you reconsider using sex as a factor, as it gives more information without need for more animals. If not for this study, for future ones.
Thanks for the advice, we will try to do a joint analysis of the parameters of males and females in the next study for the new models that we are developing
I think you misunderstood what I meant in my previous review regarding your reporting of more than one alpha value for significance in your methods section. One thing is the threshold for significance (alpha), which must be referred to in the methods, the other is the p value you obtain as a test statistic, which should be reported as it is (fir example p<0.05, p<0.01, p<0.001, p=0.034, etc.). You must only have one alpha (e.g. 0.05, 0.005 or 0.001) to determine what you consider to be significant, but you can report your actual p values, so there is no need to change them in the results, just the predefined alpha in your methods section (so please revert these to what you had previously reported, in the results section). Again, authors are advised to read on the meaning of alpha and p values. I suggest the following blog post by D. Lakens: http://daniellakens.blogspot.com/2021/11/why-p-values-should-be-interpreted-as-p.html -
We returned the values in the tables to the original version and made some correction in the text.
Thank you for thorough analysis of our work.